# Application of Steel Slag for Degraded Land Remediation

**Marina Díaz-Piloneta \*** , Francisco Ortega-Fernández, Marta Terrados-Cristos  and Jose Valeriano Álvarez-Cabal

Project Engineering Department, University of Oviedo, 33004 Oviedo, Spain; fdeasis@uniovi.es (F.O.-F.); marta.terrados@api.uniovi.es (M.T.-C.); valer@uniovi.es (J.V.Á.-C.)
* Correspondence: marina.diaz@api.uniovi.es; Tel.: +34-984-10-42-72

**Abstract:** Land degradation, and especially acidification, are global issues that need to be addressed. A common practice to correct this problem is the use of lime or chemical fertilisers that involve the extraction of raw materials. This study proposes a more sustainable alternative using Basic Oxygen Furnace (BOF) slag. BOF slag is the main waste from the steel industry that is usually accumulated in landfills, which also implies environmental impacts. In this study, a series of laboratory tests have been carried out to analyse the feasibility of using BOF slag for the reclamation of degraded land. For soil acidification, BOF slag will be analysed as a liming agent. On the other hand, the benefits slag can provide as a nutrient source will be tested. As an added value, pre-treated and untreated slag will be compared. The results of these short-time experiments show how BOF slag could be a sustainable alternative as liming agent and amendment. Its use increased the levels of some micro and macronutrients available for plant growth and improved soil quality. It could, therefore, be a sustainable management practice that makes an important contribution to the circular economy.

**Keywords:** land degradation; steel slag; soil amendment; acidification; liming agent; circular economy; nutrient source





## 1. Introduction

Land degradation is a major global problem. Soil is one of the main natural resources that supports life on Earth but the human population burden, coupled with rapid industrial expansion, has severely affected this resource [1,2]. The annual global cost of land degradation due to land use/cover change and using land degrading management practices is about USD 300 billion [3]. Land degradation can occur for different reasons. It is usually classified into physical (erosion), chemical (salinisation, acidification, fertility depletion) and biological (deforestation, rangeland degradation). Furthermore, land degradation is also related with land contamination by metallic elements.

Soil acidification is one of the main degradation problems. It is receiving increasing attention, in both non-agricultural and agricultural systems, because of its essential effects on the phytoavailability of heavy metals in the soil, the quality of the soil environment and food safety and human health [4,5]. Approximately 30% of the total land surface (ice free) is affected by soil acidification, and this percentage is gradually increasing [6,7]. Acid soils can occur naturally, or soils may slowly acidify under natural conditions over hundreds to millions of years, mainly related to weathering of minerals, soil precursor materials and rainfall [8,9]. However, human activity accelerates and causes soil acidification due to long-term fertilisation, increased absorption and removal of basic cations by crops, industrial climate conditions, urbanisation, or afforestation [10–12].

For example, chemical fertilisation is a common practice in agricultural production for higher yields and it is a major contributor to soil degradation [13]. It is one of the most important causes of soil acidification [14,15]. Regarding the application of macronutrients such as nitrogen, phosphorus and potassium, many people believe that more fertilisers will provide higher productivity. However, this leads to over-fertilisation which reduces soil quality due to acidification [16]. Furthermore, the production of chemical fertiliser is

energy intensive. About 70% of fossil energy use and $CO_2$ emission in the agricultural operating sector is associated with the production and use of chemical fertilisers [17].

Reversing acidification and land degradation therefore makes sense in light of these global trends, and the use of soil amendments is a key component of the process. The addition of amendments to degraded/disturbed soils is known to have beneficial effects and can potentially address many of the soil degradation issues [18]. Liming is a popular practice in mitigating soil acidification, and it has been used to correct soil acidity. The most commonly used liming materials are ground limestone ($CaCO_3$), dolomitic limestone ($CaMg(CO_3)_2$) and quicklime or burnt lime (CaO) [19]. However, lime application is the most cost-effective method for correcting soil acidity so its use can be limited by availability and cost [20]. Furthermore, mining and processing of these materials causes other environmental problems such as demolishing the natural ecosystem and consuming huge amounts of energy. Thus, alternative or complementary methods need to be developed [21]. Many industrial by-products and wastes can act as alternative amendments. However, their use and environmental impacts need to be properly analysed in order to avoid further harmful damage [22].

Steel slag is the main solid waste in steel industry [23]. It is obtained either by processing hot melted metal, scrap and fluxes with lime in a Basis Oxygen Furnace (BOF) or by melting scrap with high electric current in an Electric Arc Furnace (EAF). The worldwide output of steel slag is reported to be over 1600 million tons annually [24]. Around 70% of the world's steel production depends directly on the blast furnace process (integral route), as the availability of scrap limits electric arc furnace production to 30% of global demand [25]. In 2008 between 230 and 280 million tonnes of BOF slag were produced globally [26] and current slag generation is estimated to be between 200 and 250 kg per tonne of steel produced [27]. The environmental impacts of this large amount of slag generated and accumulated has prompted scientists and engineers to work on novel solutions based on more eco-friendly industrial concepts [28,29].

The characteristics and composition of BOF slag vary from site to site, as well as over different time periods at the same site due to the sources of iron ore and scrap use in the process [30]. Although the percentages may differ, these variations are not usually very significant, and the major component of BOF slag is calcium oxide among others such as iron, silica, magnesium or phosphorous [31]. Therefore, BOF slags have valuable nutrients which can be used for fertiliser applications [32]. Furthermore, these slags have high liming values, which can assist in pH buffering of soils, so they can be used as a soil amendment and liming agent and substitute natural lime products [33,34]. This can reduce the mining processes and chemical fertiliser production, as well as provide additional health benefits to soils.

Currently, there is limited knowledge of the land application and risks of slag utilisation. This lack of understanding has resulted in slag being used inappropriately, and the resource is often wasted in landfills [35]. The main application of BOF slag used to be as an aggregate in the construction sector. However, its free lime content limits its use in this field [36]. In the presence of water, free lime hydrates and results in a volume increase [37]. This swelling can lead to structural problems and limits its engineering applications. Therefore, stabilisation of slag is usually needed before its use in the construction sector. However, the effects of treating slags are unknown in the environmental field as structural problems may not be the only direct impact of such pretreatments. For this reason, this research analyses the influence of both treated and untreated slag for land applications.

This study proposes a solution to the two problems raised: soil degradation and BOF slag accumulation. To this end, a series of laboratory experiments were conducted to investigate the feasibility of using BOF slag for degraded land remediation in two different ways:

- As liming material;
- As amendment nutrient source.

Furthermore, the applications proposed in this study do not involve any structural use, so no stabilisation is required. This is an advantage over other applications, both from an environmental and economic point of view.

## 2. Materials and Methods

To achieve the presented objectives, an experimental design program for this research was developed, as shown in Figure 1. This study is divided into two different tests. On the one hand, BOF slag will be analysed as a substitute for natural lime used as a liming agent. On the other hand, the benefits it can provide as a nutrient source, with and without chemical fertilisers (NPK), will be tested.

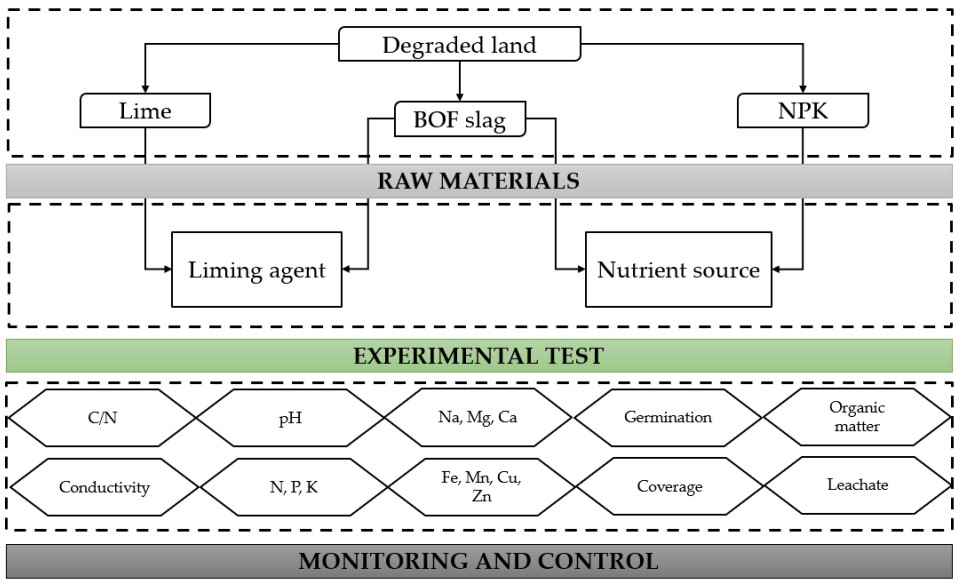

**Figure 1.** Experimental design program.

### 2.1. Basic Oxygen Furnace Slag

BOF slag is formed in the process of converting carbon-rich molten pig iron into steel. In this process, lime and dolomite are added so that slag can capture undesirable elements, such as phosphorus and silicon, and to protect the refractory lining of the furnace [38]. Once the reaction is complete, steel and slag are separated and the slag is transported to further processing facilities or directly to solidification and melting facilities, after which it is sent to a disposal or storage site [39].

Influenced by different steelmaking processes, the chemical composition of steel slags mainly consists of elements such as calcium, iron, silicon, magnesium, aluminium, manganese and phosphorus. A representative sample of BOF slag (in terms of composition and grain size) was collected from a Spanish steel production company. In the plant, after the hot molten slag is solidified and cooled, it is crushed and magnetically separated to recover the metallic iron. Then, the slag is sorted into two sizes before being disposed of in landfills. After this process, two different particle sizes are obtained, which are the ones used in the tests proposed in this paper: 0–20 and 20–50 mm. This avoids the need for preprocessing of the material and ensures its direct reuse.

The chemical composition obtained by X-ray fluorescence (XRF) is presented in Table 1. According to this, lime (CaO), silica ($SiO_2$), iron oxides (FeO) and alumina ($Al_2O_3$) are the main chemical constituents of BOF slag. As can be seen, there are no significant differences between the two particle sizes analysed.

**Table 1.** Chemical composition of BOF slag.

| % | BOF 0–20 mm | BOF 20–50 mm |
|---|---|---|
| $SiO_2$ | 13.71 | 12.77 |
| $Al_2O_3$ | 8.38 | 7.07 |
| FeO | 9.50 | 11.57 |
| $Fe_2O_3$ | 1.83 | 1.69 |
| $Fe_{total}$ | 14.38 | 19.53 |
| $P_2O_5$ | 1.22 | 1.27 |
| CaO | 45.98 | 44.37 |
| MgO | 2.97 | 2.62 |
| $K_2O$ | 0.041 | 0.051 |
| $Na_2O$ | 0.014 | 0.012 |
| MnO | 3.64 | 3.31 |
| $TiO_2$ | 0.64 | 0.59 |

BOF slag was weathered, watered and turned daily for one month [40]. Table 2 shows the free lime content for treated and untreated slag.

**Table 2.** Free lime content (%) of BOF slag before and after treatment.

| BOF Slag | 0–20 mm | 20–50 mm |
|---|---|---|
| Untreated | $8.03 \pm 0.07$ | $7.54 \pm 0.07$ |
| Treated | $4.39 \pm 0.07$ | $4.52 \pm 0.04$ |

*2.2. Characteristics of Degraded Soil*

The soil used in the tests was collected from a degraded area in the North of Spain (43°19′46.8″ N, 4°48′18.2″ W) from the top 20 cm. In addition to the environment, the main cause of soil degradation and acidification is humans' actions through poor livestock and agricultural practices. The main soil characteristics are shown in Table 3 classified according to Landon [41]. The soil pH and electrical conductivity were measured in a 1:2.5 and 1:5 soil/water suspension, respectively. Extractable K, Ca and Mg contents were determined by extraction with ammonium acetate [42]. The loss on ignition (LOI) method was used to estimate the soil organic matter [43]. The phosphorus determination was carried out using the Olsen method [44].

**Table 3.** Main soil characteristics.

| Parameter | Unit | Value | Classification |
|---|---|---|---|
| Clay | % | 9.90 | - |
| Silt | % | 19.80 | - |
| Sand | % | 70.30 | - |
| pH | - | 4.20 | Very low |
| Conductivity | $\mu S\ cm^{-1}$ | <70 | Very low |
| Organic matter | % | 2.03 | Normal |
| Nitrogen (N) | $mg\ kg^{-1}$ | 752 | Low |
| Phosphorus (P) | $mg\ kg^{-1}$ | 9.80 | Low |
| Potassium (K) | $meq\ 100g^{-1}$ | 0.08 | Very low |
| Calcium (Ca) | $meq\ 100g^{-1}$ | 1.48 | Very low |
| Magnesium (Mg) | $meq\ 100g^{-1}$ | 0.24 | Very low |
| C/N | - | 15.60 | High |

The soil can be classified as a sandy loam texture with acid pH (4.2) and low fertility with a high liming demand, low N contents, negligible K and low content of P [41]. All other micronutrients are also at low levels. The high C/N ratio, coupled with other factors (low pH, insufficient phosphate and low conductivity) indicates a poor ability to produce nitrate.

*2.3. Test 1: BOF Slag as Liming Agent*

Liming improves the physical properties of soil through increasing base cations [45]. Calcium and magnesium cations can bind to organic matter as well as mineral colloids. Because of the beneficial influence of liming on soil structure, there has been much research on the use of acid-neutralising materials for improving degraded soils [46]. Some slags are used as liming materials to adjust soil pH and soil conductivity and to increase base saturation and cation exchange capacity [47]. BOF slag offers a range of environmental benefits, including being used as liming material which can substitute natural lime products. Therefore, this first test is proposed to analyse the technical feasibility of BOF slag for liming instead of natural lime.

Throughout Europe, each country has its own specifications for liming materials, but the European Union has proposed harmonising regulations. Regulation (EU) 2019/1009 adds liming materials to the European Fertiliser Regulations so that they can be sold as 'Liming Materials' [48]. For all materials with liming value, two important quality characteristics are [49]: (i) the neutralising value (NV) and (ii) the particle size. Studies evaluating limestone particle size show that the finest material is best for increasing soil pH and reducing the concentration of exchangeable Al [50]. To analyse the influence of slag particle size, this test used three BOF slag sizes and thus three liming types: 0–20, 20–50 mm and a mix of both (all-one). The last one (all-one) was carried out in a 50–50 ratio of 0–20 mm and 20–50 mm.

The liming material required to adjust the pH of soil varies depending on the soil type, and each country sets its own recommendations. For example, Table 4 shows the quicklime (CaO) required depending on the type of soil and pH according to the United Kingdom *Fertiliser Manual (RB209)* [51].

**Table 4.** Quicklime (CaO) required (g m$^{-2}$) depending on the soil type and initial pH.

| Soil Type | pH = 5.0 | | pH = 5.5 | | pH = 6.0 | | pH = 6.2 | |
|---|---|---|---|---|---|---|---|---|
| | Grass | Arable | Grass | Arable | Grass | Arable | Grass | Arable |
| Organic soils | 600 | 1600 | 0 | 800 | 0 | 0 | 0 | 0 |
| Sands and loamy sands | 500 | 1000 | 300 | 700 | 0 | 400 | 0 | 300 |
| Sandy loams and sit loams | 600 | 1200 | 400 | 800 | 0 | 500 | 0 | 400 |
| Clay loams and clays | 700 | 1400 | 400 | 1000 | 0 | 600 | 0 | 400 |

Less liming material is required for non-agricultural land. The aim of this study was not to prepare the land for agriculture but to restore the soil health by providing a homogeneous cover and to correct its acidity. Therefore, according to experts, 450 g of quicklime was set as the required amount of material per square metre. The BOF slag used has a lower percentage of CaO, which varies according to size (Table 1). The lime used is not 100% CaO either, so it is necessary to adjust the rate required according to the CaO concentration of each material (Table 5). Soil was mixed with the different liming materials and disposed of in pots.

**Table 5.** Rate of liming material per m$^2$ and per pot used in the test.

| Material | % CaO | g m$^{-2}$ | g pot$^{-1}$ |
|---|---|---|---|
| BOF slag 0–20 mm | 50% | 900 | 77.6 |
| BOF slag 20–50 mm | 42% | 1071.5 | 92.4 |
| BOF slag all-one | 46% | 978.3 | 84.4 |
| Lime | 77% | 584.4 | 50.4 |

Soil with lime treatment was also included in the experimental design. After that, the seed mixture (Table 6) was placed in the pots. Three replicates per treatment were prepared. The samples were placed in a laboratory, watered and monitored during the whole length of the trial (17 weeks).

**Table 6.** Seed mixture used in the test.

| Family | Species | Common Name | % |
|---|---|---|---|
| *Fabacae* | *Trifolium repens* | White clover | 5 |
| | *Medicago sativa* | Alfalfa | 5 |
| | *Trifolium pratense* | Red clover | 5 |
| *Poaceae* | *Lolium perenne* | English ryegrass | 20 |
| | *Festuca rubra* | Red fescue | 15 |
| | *Festuca arundinacea* | Tall fescue | 15 |
| | *Lolium multiflorum* | Italian ryegrass | 13 |
| | *Dactylis glomerata* | Cat grass | 13 |
| | *Agrostis tenuis* | Colonial bent | 5 |
| | *Holcus lanatus* | Tufted grass | 2 |
| *Plantaginaceae* | *Plantago lanceolata* | Ribwort plantain | 2 |

Canopy cover, germination rate, leachate pH and plant height were monitored every week. Plant height was measured in a forced manner and in three different points. Plants coverage rate was measured using *Canopeo. Canopeo* is a mobile application that allows the monitoring of plants growth by calculating the fraction of green plant cover [52]. Its operation is based on colorimetry, offering much more accurate in situ results than common visual examinations [53]. The pots were provided with holes to allow water leaching after irrigation. Leachate was collected in trays every week, and their pH was measured with a pH meter (S47 SevenMulti). After 14 weeks, plants were harvested by cutting at the soil surface and oven dried (at 70 °C for 48 h) to record the shoot dry biomass.

### 2.4. Test 2: Soil Amendment and Nutrient Source

Soil pH significantly affects the availability and uptake of micronutrients by plants. Ca, Mg and K have a direct relationship with pH, as in acid soils they decrease quickly. BOF slag contains not only Ca and Mg (Table 1), which enables its use as a liming material, but also P, Mn, Fe and other elements with fertilising effects.

Phosphorus (P) is one of the essential elements for plant life. Adequate levels of P in the soil are important for root development and plant growth [54]. Manganese (Mn) is known to influence chlorophyll production and to support photosynthesis [55]. Iron (Fe) transforms hydrogen sulphide in the soil into iron sulphide, thus rendering it harmless and reducing damage to plant roots [56].

The same soil used in the first test was used in this second one to analyse the influence of BOF slag as a soil amendment and nutrient source in degraded soils. Five amendment materials were analysed: quicklime (L); $N:P_2O_5:K_2O$ at 15:15:15 (NPK); BOF slag 0-20 mm (E); all-one BOF slag (T) and all-one treated BOF slag (S). Untreated soil control (C) was also included in the experimental design. These materials were used to make various mixtures with different compositions. In this case, BOF slag was applied mixed with the soil in proportions of 25% (E25, T25, S25) and 50% (E50, T50, S50) of the pot's volume. Quicklime was applied in the same rate than in the previous test. The mixtures were made with and without fertiliser (NPK) applying a dose of 12 g m$^{-2}$. The seeds used in this test are shown in Table 6. Three replicates per mixture were prepared. The pots were placed in a laboratory, watered and monitored for 12 weeks.

Plant height, germination rate and plant coverage were measured every week following the methodology described in the previous section. After 5 and 10 weeks, dry biomass was recorded as in Test 1. All the soil amendments were characterised at the end of the test as described in Section 2.2 for the soil. For the extraction of micronutrients (Fe, Mn, Cu and Zn), diethylenetriaminepentaacetic acid (DTPA) was used as a chelating agent [57]. At the end of the test, the change over the initial pH was analysed. The soil pH was measured in a 1:2.5 soil/water suspension. The leachates were collected at the end of the test to measure the concentration of metals (Cr, Pb, V, Co, Cu, Sr, Al, Ni, As, Zn, Hg, Cd and Mn) as a quality control procedure. The chemical composition of the leachate from each pot was

also analysed with inductively coupled plasma mass spectrometry (Agilent 5975C Inert XL MSD).

### 2.5. Statiscal Analysis

The recorded data were statistically analysed using SPSS 22.0 and MS Excel 2016. Student's *t*-test for independent samples was employed to examine the statistical significant difference among the mean of the application rates at a level of $p < 0.05$.

## 3. Results and Discussion

### 3.1. Liming Agent

The germination period lasted for the first five weeks. The performance of all samples was quite similar. No notable differences were obtained in terms of germination index, height and canopy cover. However, there were clear performance differences in the pH of the leachate (Figure 2).

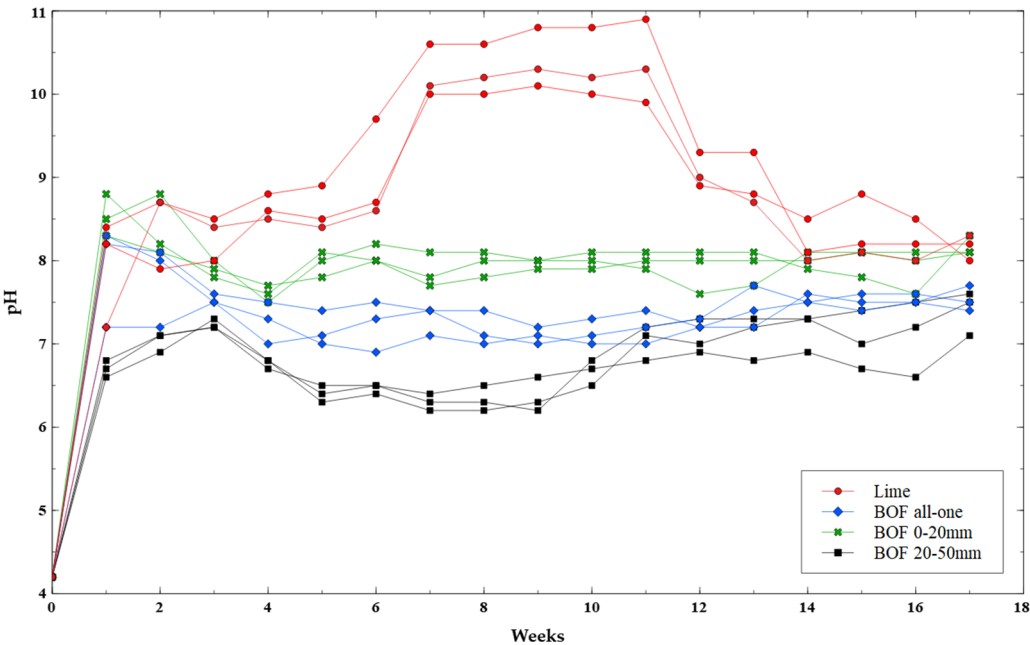

**Figure 2.** Changes in pH leachate of different liming materials.

Changes in pH reflect different buffering rates. The pH levels obtained at the end of the test (week 17) varied with the particle size. Samples with 20–50 mm BOF slag were closer to a neutral pH, and the finest granulometries (0–20 mm) reached the highest values. An intermediate solution was the mix of both granulometries (all-one). There were no significant differences between samples with 0–20 mm slag and samples with lime ($p > 0.05$). However, pots with all-one and 20–50 mm slag achieved significantly lower pH levels ($p < 0.01$).

On the other hand, the finer sizes had a faster buffering effect. In the first two weeks samples reached a pH between 8–9 and during the test they had a very small variation. However, coarser granulometries offered slower and more progressive results. This shows how the finer sizes have more capacity to release the basic cations in less time.

It is interesting to analyse the performance of lime pots that reached the highest pH levels. All BOF slag samples had an initial peak that decreased after 2–3 weeks. However, the lime pots continued to increase until week 11, when they decreased to a pH of 8–9. This could imply that the effect of lime decreases faster over time, while the slag remains more stable.

The analysis of the dry biomass after 14 weeks obtained after cutting and drying each sample also shows clear differences (Figure 3). The pots with lime had the lowest

percentages and the mixes with BOF slag with a particle size of 0–20 mm had the highest values. Once again, an intermediate solution was slag with all-one grain size.

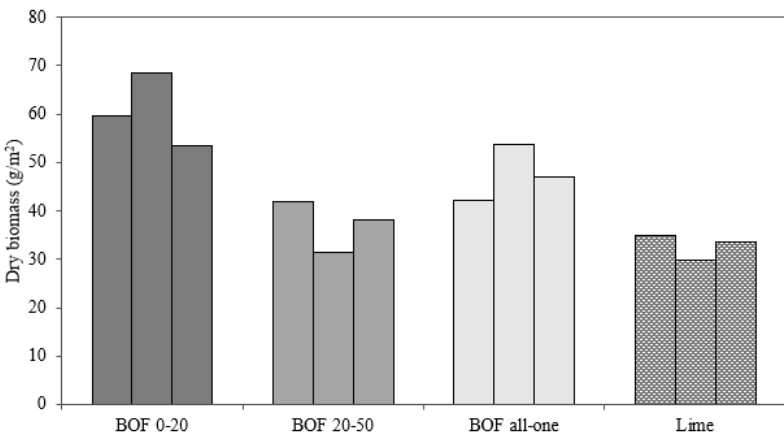

**Figure 3.** Dry biomass of all samples (g m$^{-2}$).

Samples with 0–20 mm slag significantly increase dry biomass in comparison with lime samples ($p < 0.02$). This difference is not as relevant in the rest of the granulometries ($p > 0.05$). The results show that slag can replace the lime commonly used as a liming material. Specifically, the slag with a particle size of 0–20 mm obtained a more stable buffering effect and in less time. In addition, all samples with slag achieved higher percentages of dry biomass. On the other hand, it can be seen how the buffering effect and speed varies depending on the material used and its grain size. Therefore, according to each case needs, a different type of slag with a different particle size can be applied.

In terms of the species identified after cutting, there were no appreciable differences between the granulometries of slag and lime. The most developed species was *Lolium perenne*. To a lesser extent, the growth of some leguminous types was also observed, particularly in *Trifolium repens* and *Medicago sativa*. However, competition from the tallest and most abundant species (*Lolium perenne*) made them develop more slowly. Finally, some shoots of *Lolium multiflorum* could also be identified.

*3.2. Soil Amendment and Nutrient Source*

This second test analyses the influence of BOF slag as a soil amendment and nutrient source in degraded land. In this case the plants were established after 3.5 weeks, but the germination rate was different for each sample (Figure 4).

Samples with slag have a significantly higher germination rate ($p < 0.02$). The mixtures with the most successful results both with and without fertiliser were the pots with 25% all-one slag (T25). Amendment with fertiliser showed a more pronounced initial peak. In samples without NPK, germination was more spaced out. However, the addition of fertiliser only had a significant impact on pots with lime (L) and natural soil (C) ($p < 0.01$). The worst performances in both cases were the mixtures with 50% untreated 0–20 mm slag (E50).

Results show that as the percentage of slag increases, germination is significantly reduced ($p < 0.01$). There is no appreciable difference between treated and untreated slag ($p > 0.05$) nor between untreated slag and lime. However, treated slag obtained significantly higher germination percentages than samples with lime. In addition, the finer-grained slag performed worse than the all-one samples, with and without fertiliser ($p < 0.01$).

The macronutrients available at the end of the test are shown in Figure 5 in comparison with initial soil characteristics (C0). The P content was higher in amendments with slag ($p < 0.02$) and all samples with fertiliser addition ($p < 0.01$). However, there was no difference between treated and untreated slag, nor between slag granulometries.

In contrast, the concentration of K in plants decreased as the quantity of slag applied was increased. This effect may be due to a dilution effect for P and also to increased

competition with Ca and Mg cations introduced by slag [58]. However, this effect is apparently corrected when mixing the slag with fertiliser.

The addition of slag implies a significant reduction in nitrogen levels ($p < 0.01$) and no differences were seen with the presence of NPK ($p > 0.05$). Samples with 25% slag achieved significantly higher nitrogen levels ($p < 0.01$) than pots with 50%, so it can be concluded that the nitrogen concentration decreases as the amount of slag increases.

The effect of slag treatment on Ca concentrations in plants was clear. Calcium increased significantly in mixtures with slag amendments ($p < 0.01$) and this rate was higher in untreated slag samples ($p < 0.02$). This may be because the treated slag has a lower percentage of free lime. In presence of water, free lime hydrates and forms portlandite ($Ca(OH)_2$) [59]. Portlandite has a lower density than CaO, so the hydration of free CaO results in a volume increase [37].

There was no significant difference between amendments with NPK ($p > 0.05$). However, as can be seen in the figure, samples with fertiliser had lower Ca values. This lower concentration of Ca could be due to the larger quantities of other cations, such as K competing with Ca and Mg cations [60]. Nevertheless, this is not the case for the pots with 0–20 mm slag since amendments E25 and E50 had extremely high levels of calcium both with and without fertiliser.

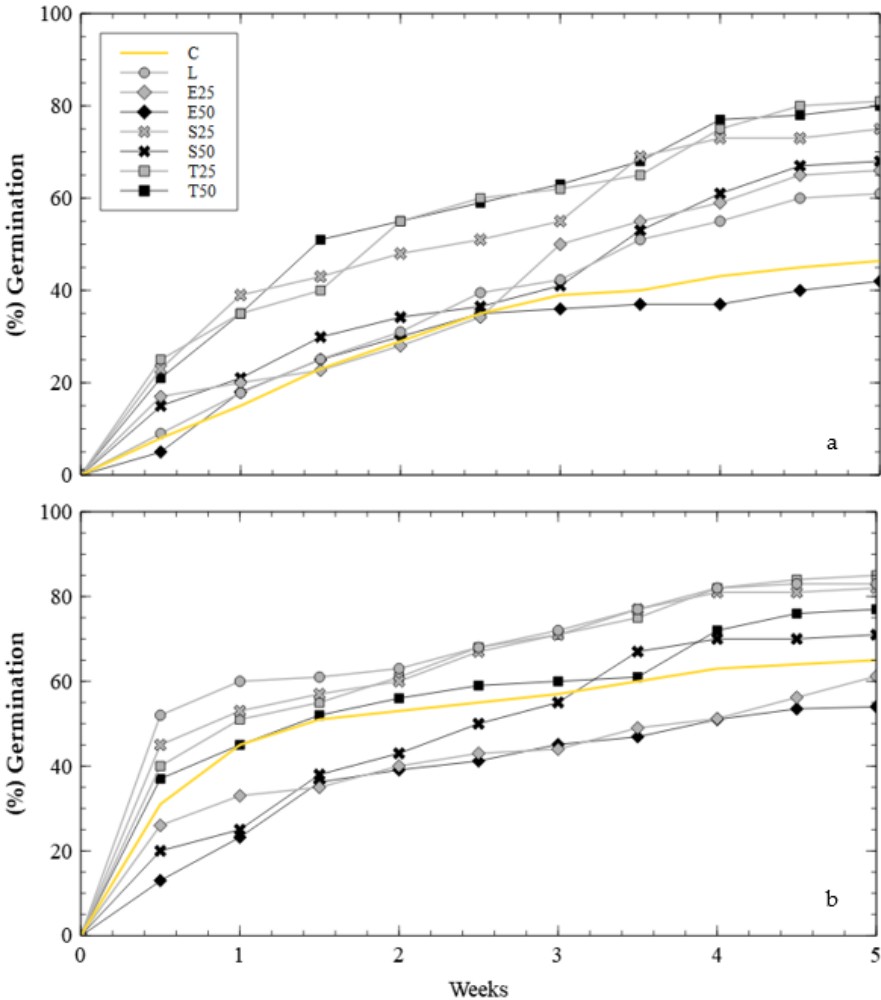

**Figure 4.** Germination rate: (**a**) samples without NPK; (**b**) samples with NPK.

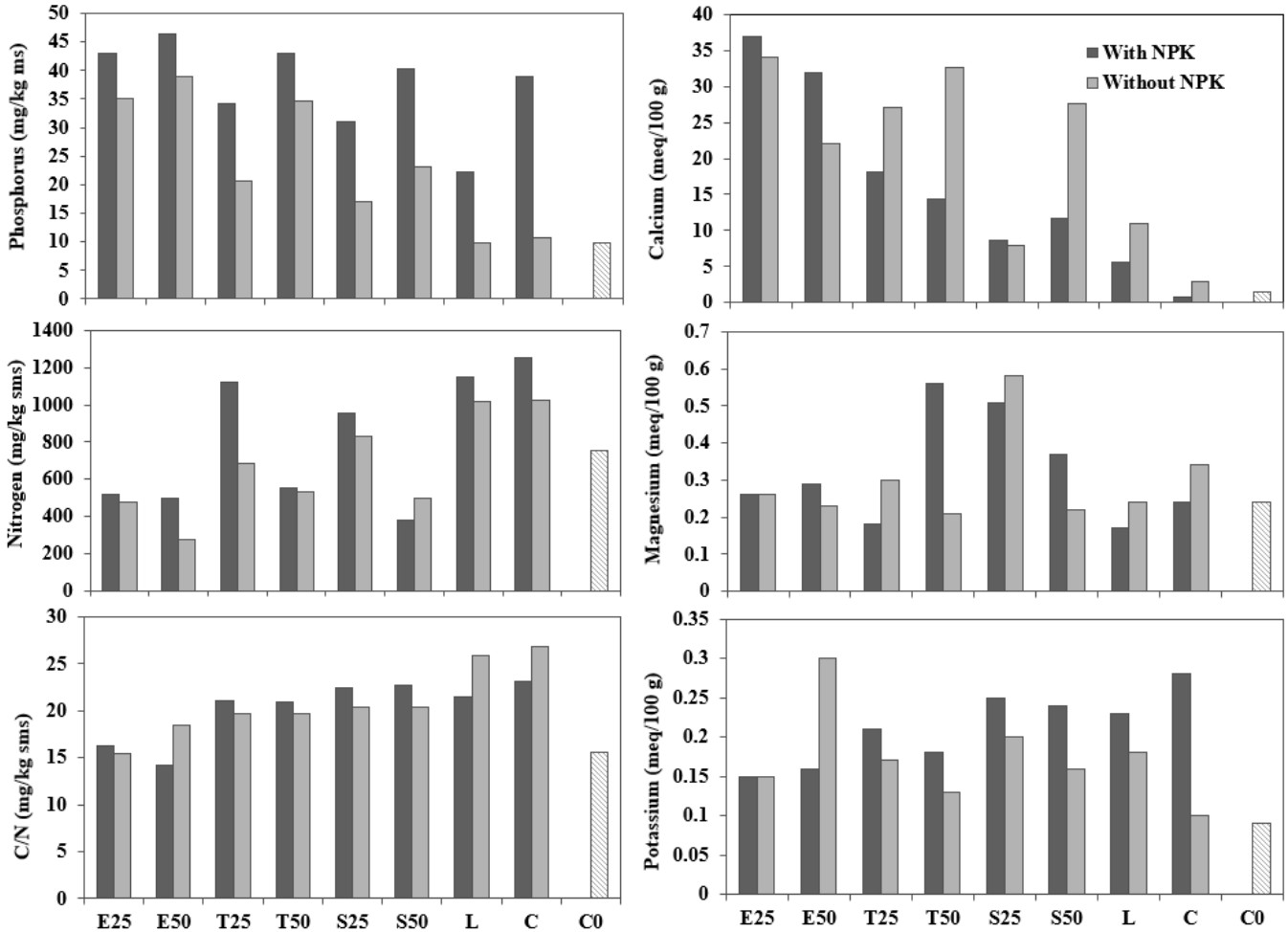

**Figure 5.** Macronutrient concentration and soil parameters in comparison with initial soil characteristics (C0).

The presence of Mg in the soil was significantly higher in pots with treated slag ($p < 0.05$). However, the higher amount of calcium provided by BOF slag created a clear competition between Ca and Mg and led to a decrease in Mg uptake. Samples with treated slag (S25, S50) reflecting lower levels of calcium reached higher levels of magnesium.

In addition to the macronutrients, it is also necessary to analyse the micronutrients required in small, but critical, quantities for the normal healthy growth of plants (Table 7). Iron (Fe), copper (Cu), manganese (Mn) and zinc (Zn) are heavy metals essential for plants. Critical deficiency concentrations of these micronutrient heavy metals in most plant species [61] are generally in the range (mg kg$^{-1}$): Cu 3–5; Fe < 50; Mn 10–20 and Zn 15–20. On the other hand, toxic concentrations (in mg kg$^{-1}$) are in the range: Cu 20–100; Fe > 1000; Mn 300–500 and Zn 100–400.

In the initial soil (C0), there was a clear deficiency of all the necessary micronutrients. Cu levels did not significantly improve with the addition of slag or fertiliser ($p > 0.05$). However, samples with a higher percentage of slag (50%) significantly increased their Cu concentrations compared to samples with 25%. Neither lime nor fertiliser was sufficient to enhance Mn levels. The same applies with E25 and E50 amendments with 0–20 mm slag. However, all samples with all-one slag reached normal levels of this heavy metal and considerably increased their Mn concentration ($p < 0.01$).

**Table 7.** Effect of amendments treatments on available micronutrients (mg kg$^{-1}$) and pH.

| | Amendment | Fertiliser | Fe * | Mn * | Cu * | Zn * | pH |
|---|---|---|---|---|---|---|---|
| E25 | 25% 0–20 mm BOF slag | With NPK | 97.50 | 12.40 | 0.20 | 1.11 | 10.20 |
| | | Without NPK | 118.00 | 6.80 | 0.20 | 0.20 | 10.10 |
| E50 | 50% 0–20 mm BOF slag | With NPK | 149.00 | 1.12 | 0.23 | 0.20 | 11.40 |
| | | Without NPK | 175.00 | 1.00 | 0.33 | 0.20 | 11.20 |
| T25 | 25% all-one BOF slag | With NPK | 60.60 | 13.31 | 0.20 | 6.35 | 8.31 |
| | | Without NPK | 79.40 | 24.80 | 0.20 | 0.85 | 7.92 |
| T50 | 50% all-one BOF slag | With NPK | 63.60 | 33.40 | 0.27 | 2.01 | 8.80 |
| | | Without NPK | 79.30 | 39.30 | 0.29 | 2.08 | 9.30 |
| S25 | 25% treated BOF slag | With NPK | 50.00 | 16.48 | 0.20 | 2.06 | 7.590 |
| | | Without NPK | 63.00 | 17.39 | 0.20 | 2.10 | 7.71 |
| S50 | 50% treated BOF slag | With NPK | 49.20 | 26.40 | 0.23 | 1.23 | 8.87 |
| | | Without NPK | 60.40 | 14.90 | 0.20 | 1.96 | 8.32 |
| L | Lime | With NPK | 74.00 | 1.00 | 0.20 | 2.62 | 8.12 |
| | | Without NPK | 87.00 | 1.09 | 0.20 | 1.76 | 8.15 |
| C | Soil control | With NPK | 48.20 | 2.81 | 0.26 | 6.78 | 4.46 |
| | | Without NPK | 44.70 | 5.28 | 0.20 | 6.54 | 4.30 |
| C0 | Initial soil | - | 51.00 | 1.00 | 0.29 | 1.72 | 4.20 |

* All values in mg kg$^{-1}$.

Zinc levels in the soil were significantly higher in control samples (C). Again, a clear difference can be seen between pots with 0–20 mm slag and all-one size. In this case, the samples with all-one slag reached significantly higher concentrations ($p < 0.05$). Regarding iron levels, all mixtures achieved normal values except from those of natural soil with and without fertiliser (C). Amendments E25 and E50 with fine granulometry reached notably higher levels ($p < 0.01$) than all-one slag, treated (S) and untreated (T).

The pH increased significantly in the samples with slag. The results obtained in the previous test can be checked again here. Samples with 0–20 mm slag reached the highest levels, in this case significantly higher than lime pots ($p < 0.01$). Again, in treated slag amendments, this increase was significantly lower than in untreated slag ($p < 0.02$), probably due to the lower amount of free lime.

Figure 6 shows the coverage rate and dry biomass of samples after 5 weeks (first cut) and 10 weeks (second cut). Error bars represent standard deviation of triplicate measurements.

The results showed a direct relationship between the coverage ratio and the dry biomass. Amendments with all-one slag obtained the best results, followed by mixtures with treated slag and lime. These last ones performed better with fertiliser, but all-one slag samples improved without it. T50 without NPK and S25 with fertiliser had similar performance. However, after the first cut, T50 decreased its productivity.

A clear variation can be seen between the first and the second cut. While for the first cut there were no significant differences between slag amendments and the rest of the mixes ($p > 0.05$), in the second one the dry biomass decreased significantly in pots without slag ($p < 0.02$). Furthermore, there were no significant differences between the cuts for amendments with 25% slag ($p > 0.05$). Samples with treated slag and all-one sizes reached acceptable growth levels. However, amendments with higher slag content worsened after the second cut.

All pots with 0–20 mm slag had significantly lower dry matter percentages than all-one samples in both cases. However, the particle size did not influence the coverage ratio ($p > 0.05$). There were also no differences between treated and untreated slag ($p > 0.05$). The results show that the addition of fertiliser and lime is not enough for soil improvement, as the L and C samples reached very low growth levels after the second cut. Nevertheless, in combination with slag, very good yields were obtained.

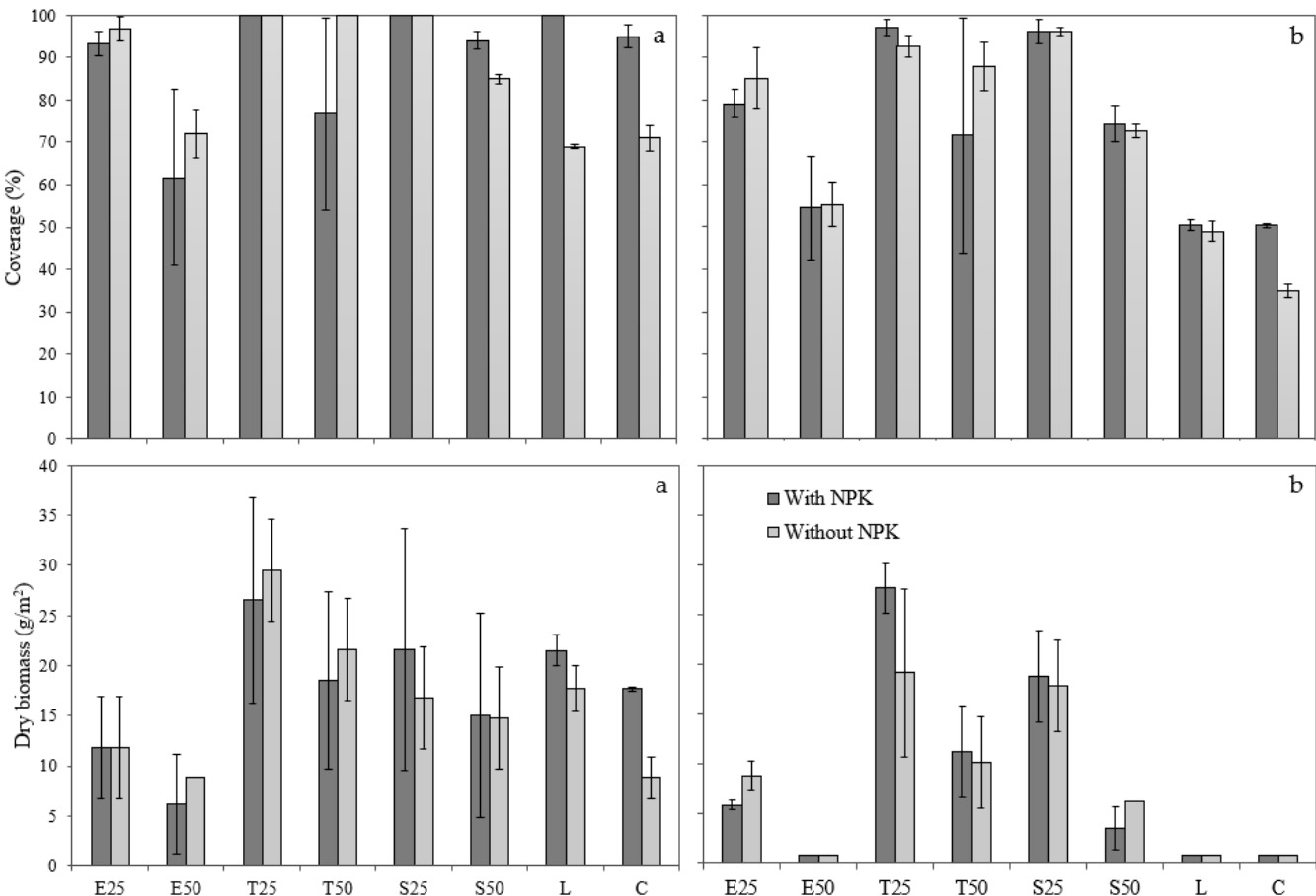

**Figure 6.** Coverage rate (%) and dry biomass (g m$^{-2}$): (**a**) first cut after 5 weeks; (**b**) second cut after 10 weeks.

Despite the similar germination rate in T25 samples, with and without fertiliser, the results showed that over time no clear establishment was achieved in the case of adding fertiliser since the variability of these samples was very high. Generally, samples with slag had higher deviations. It would therefore be interesting to analyse these results again with a higher number of repetitions.

Regarding the species identified in the first cut, the most common one in all samples was *Lolium perenne* and some unidentified *Trifolium* types (*Trifolium repens* or *patrense*) as the distinctive characteristics were not fully developed. The majority presence of *Lolium perenne* was not surprising, as it had the highest percentage in the seed mixture used. The proliferation of *Trifolium* was probably due to the increase in pH, as this species has a certain affinity for more basic pH values [62]. The *Trifolium* content was visible in both the fertilised and unfertilised samples. However, it should be noted that the increase in this species was more striking in the case of the fertilised samples due to the higher P levels.

The visual perception after the first cutting was positive, as it had the appearance of a normal grassland except for the control samples with natural soil without fertiliser. In the second cut, the species that were previously under strong competition grew more homogeneously, creating a stratigraphic equality between *Lolium perenne*, *Trifolium* and *Plantago lanceolata*. The appearance of *Plantago* after the first cut can also be explained by the affinity of this species for calcium [63]. The occurrence of species is not influenced by the type of amendment, only by the addition or not of fertiliser in the case of *Trifolium*.

Table 8 shows the heavy metals analysed from leachate collected from each sample at the end of the test. The results highlighted in bold represent the values exceeding thresholds.

**Table 8.** Heavy metals in leachates (mg L$^{-1}$).

| Amendment | Fertiliser | Cr | V | Cu | Sr | Al | Ni | As | Zn |
|---|---|---|---|---|---|---|---|---|---|
| E25 | With NPK | 0.0023 | 0.0105 | 0.0988 | 0.4239 | 0.2785 | 0.0092 | 0.0507 | 0.0332 |
|  | Without NPK | 0.0022 | 0.0398 | 0.0454 | 0.2180 | 0.2700 | 0.0075 | 0.0452 | 0.0499 |
| E50 | With NPK | <0.002 | 0.0022 | 0.0818 | 0.7164 | 0.0799 | 0.0038 | 0.0377 | 0.0186 |
|  | Without NPK | <0.002 | <0.002 | 0.0976 | 2.3522 | 0.0579 | 0.0044 | 0.0389 | 0.0076 |
| T25 | With NPK | 0.004 | 0.0286 | 0.0321 | 0.1485 | 0.2532 | 0.0073 | **0.0766** | 0.0514 |
|  | Without NPK | <0.002 | 0.022 | 0.007 | 0.1470 | 0.0311 | <0.002 | 0.0124 | 0.1292 |
| T50 | With NPK | 0.0032 | 0.0037 | 0.0769 | 0.3636 | 0.2157 | 0.0081 | **0.0714** | 0.0569 |
|  | Without NPK | 0.0039 | 0.0112 | 0.0798 | 0.5984 | 0.3726 | 0.0086 | **0.0935** | 0.114 |
| S25 | With NPK | 0.008 | **0.1428** | 0.0138 | 0.2005 | 0.1614 | 0.0042 | 0.0517 | 0.1561 |
|  | Without NPK | <0.002 | 0.0204 | 0.0063 | 0.1656 | 0.0335 | <0.002 | 0.0149 | 0.1833 |
| S50 | With NPK | 0.0031 | 0.0935 | 0.0176 | 0.1425 | 0.0487 | 0.0033 | 0.0227 | 0.2728 |
|  | Without NPK | 0.0025 | 0.0666 | 0.0135 | 0.1282 | 0.0461 | <0.002 | 0.0137 | 0.2172 |
| L | With NPK | 0.0062 | 0.0145 | 0.0043 | 0.6952 | 3.4287 | 0.0073 | <0.002 | 0.5077 |
|  | Without NPK | 0.0044 | 0.0053 | 0.0063 | 0.1399 | 1.0978 | <0.002 | <0.002 | **6.8505** |
| C | With NPK | <0.002 | <0.002 | 0.0051 | 0.0056 | 0.5178 | <0.002 | 0.0095 | 0.1967 |
|  | Without NPK | <0.002 | 0.0028 | 0.0118 | 0.2415 | 0.4344 | 0.0037 | <0.002 | 0.1622 |
| *FAO* | - | | *0.100* | *0.100* | *0.200* | - | *5.000* | *0.200* | *0.100* | *2.000* |
| *EU (2003/33/EC)* | - | | *0.100* | - | *0.600* | - | - | *0.120* | *0.060* | *1.200* |

No toxic concentrations of any metal were found according with FAO recommendations [64] for trace elements for agricultural water quality and EU thresholds for heavy metals in landfill leachate [65]. The levels of cobalt, mercury, cadmium, manganese and lead were negligible (<0.002 mg L$^{-1}$), so they are not represented in the table. It can be seen that Zn concentrations were higher in samples with lime. There were variations between treated and untreated slag ($p < 0.02$). Overall, samples with treated slag had higher levels of heavy metals (especially vanadium). There were no differences between slag amendments and soil for the levels of chromium, vanadium and arsenic ($p > 0.05$). However, mixtures with slag achieved significantly lower Al values ($p < 0.05$).

## 4. Conclusions

The results obtained in this study showed that BOF slag can be used as a substitute for natural lime and as amendment and a nutrient source for degraded soils improvement. On one hand, the slag with a particle size of 0–20 mm obtained a more stable buffering effect, and in less time, than natural lime. No significant differences were found in terms of pH levels achieved ($p > 0.05$) and higher percentages of dry biomass were obtained ($p < 0.02$). Furthermore, depending on the needs of each case, a different type of slag with a different particle size can be applied with different buffering effects.

In the case of nutrient source, common practices with natural materials such as lime or chemical fertilisers are not enough to improve soil quality. Phosphorus levels increased in slag samples, but potassium and nitrogen concentrations decreased. The effect on Ca levels was the most significant ($p < 0.01$), being higher in the pots with untreated slag ($p < 0.02$). The addition of slag also improved dry biomass. While the percentage of cover was not influenced by the slag particle size, amendments with 0–20 mm obtained significantly lower dry biomass percentages.

Overall, all-one slag, both pre-treated and untreated, improves soil quality and macro- and micronutrients available for plant growth. This is not the case for fine-grained slag (0–20 mm), which has been considered unsuitable for this type of activity. The best-performing amendments were T25 and S25. As the slag content increases, the mobility of some basic nutrients such as Mg decreases, so an addition of more than 25% is not recommended.

Samples with slag usually showed higher deviation in the results, so in future it would be useful to repeat the test with a higher number of replicates. In addition, it would also be interesting to analyse the heavy metals in soil leachates before and after the test to see how they vary over time. Regarding this, it would be very interesting to carry out a long-term study on a real scale and the heavy metal content of the slag, in order to verify that the use of this waste is not harmful to the environment.

The use of BOF slag for remediation of land degradation could be a sustainable management practice as it recovers waste that would normally accumulate in landfills and also avoids the extraction of raw materials. This contributes to the circular economy and to the reduction of factors responsible for climate change.

**Author Contributions:** Conceptualisation, J.V.Á.-C.; methodology, F.O.-F. and M.D.-P.; formal analysis, M.T.-C. and M.D.-P.; investigation J.V.Á.-C. and M.D.-P.; resources, F.O.-F.; writing—original draft preparation, M.D.-P. and M.T.-C.; writing—review and editing, J.V.Á.-C. and F.O.-F.; visualisation, M.T.-C.; supervision, J.V.Á.-C. and F.O.-F.; project administration F.O.-F.; funding acquisition, F.O.-F. and M.D.-P. All authors have read and agreed to the published version of the manuscript.

**Funding:** The results of this research was funded by Ministry of the Economy, Industry and Competitiveness and the State Research Agency of Spain, grant number RTC-2017-6329-5, and this paper was funded by Regional Ministry of Science, Innovation and University of the Principality of Asturias grant number AYUD/2021/50953.

**Data Availability Statement:** Not applicable.

**Conflicts of Interest:** The authors declare no conflict of interest.

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
