# Peer review of "Application of Steel Slag for Degraded Land Remediation"

_land, doi:10.3390/land11020224_

Round 1

Reviewer 1 Report

I really enjoy reading and reviewing the paper, this only small suggestions which should be included to improve the paper are listed below:

Line 9 – „new” remove

Line 71 – this variations are really slight due to norms and requirements that need to be fulfilled in case of both input materials and output product. Search for the literature about it

75 – phosphorus is a nutrient, the others…well Mg is very limited and listed as critical so as P. And to be used for fertilization also requirements and norms must be fulfilled including content of e.g heavy or toxic elements

83 – unnecessary “dot” appeared there

88-92  -first time I saw in a paper this kind of “table of content” I suggest to remove it unless the LAND requires differently

110 – disposed instead of disposal

119 – to use were? for building industry it is a problem for land applications swelling is not so important. Maybe restructure a bit this paragraph, because the following info appears there but later than expect. Anyway this is the material characterization chapter and this paragraph is very general so I suggest to move it to the discussion part

136 –pH how much? The same problem as in the paragraph mentioned before. Rephrase it a bit

191 –what was the length of experiment?

221 – why not at the end of experiment (12 weeks)?

231 – this sentence I do not understand

Fig.5 use dost instead of commas on the scale. This fig.is not very sharp so if can be improved please do

Table 7  - explain the symbols directly next to the table or place the table below the text where it is describe

330 – either I am lost here or the coverage rate was not described

366  - I missed the description of leaching? How was it performed? And also the pH of soils? Was it measured in suspension after "watering" ?

Reviewer 2 Report

The manuscript titled “Application of steel slag for degraded land remediation” scopes mainly on the use of slags as soil additive (source of micronutrients, change the pH of soils). Specific comments are presented below.

Line 9: "BOF slag" - B O F letters should be explained in parentheses (Basic Oxygen Furnace)

Line 13: "degraded land"-It should be clearly stated which type of degradation in your study.

Lines 14-15: I think that it should be mentioned that exposure of slags on environmental factors can be toxic in a long time scale even, if you have steel slags. Your slags are enriched in Al - the element could be toxic for plants (especially for roots). You have also mentioned that use of slags increases the content of microelements. I did not see the results of metallic elements in slags i.e., Cu, Zn, Pb. Do you have such results? You have results of metallic elements content in leachates.

Lines 25-27 and Introduction: Land degradation is also related with land contamination by metallic elements (i.e. from deposited slags and wastes from thermal power stations and households; emission and imission of particulate matter from smelters, thermal power stations, and households; see:

https://www.sciencedirect.com/science/article/abs/pii/S0956053X21003263

Such wastes can also acidify the soil (depending on S content). On the other hand, ashes can be the source of Ca for soils. It should be introduced in this part.

Line 58: "and sulphate" has to be removed from the sentence. Sulphates can negatively affects the soil (pozzolanic process, to high content of S).

Lines 88-90: This part has to be removed from the manuscript. Not necessary.

Line 109: Could you add the name of company?

Line 112: Are the slags milled into two fractions in the company? I think that "hot"/melted slag is poured out. Therefore, relatively large fragments of slags are after cooling out.

Do you have results of mineralogy of BOF slag (i.e. measured using XRD)? I think that it is important to indicate phase composition of slag. You can discuss susceptibility of phases to weathering.

Lines 119-125: This part should be added in discussion.

Line 135: It should be precisely stated where soil was collected.

Table 3. You have mistake in soil texture groups. It should be "Silt" instead "Sint". For the classification (Normal, Very Low etc.), the reference is needed.

Line 146: Phrase "a light-textured" has to be removed. Please, use soil texture classification according USDA or FAO WRB

https://www.nrcs.usda.gov/wps/portal/nrcs/detail/soils/survey/?cid=nrcs142p2_054167

Table 6: Latin names of plant species should be written in italic.

Line 196: Please, add the name of pH-meter.

Line 224: Citation is needed.

Line 228: Please, add the name of ICP-MS.

Line 256: In my opinion, the word “perfectly” is to strong. As you mentioned, slags have to be processed before use in arable soil. So, the process is also cost-intensive.

Line 291: Citation is needed.

Line 299: Citation is needed.

Line 356: Citation is needed.

Line 364: Citation is needed.

Line 370: “No toxic concentrations of any metal were found.” Please, add the citation with the name of classification of toxic/no-toxic leachates.

Line 376: Please, use “may be” instead “can be”. In my opinion each slag can be toxic for plants in a long-time scale. You do not have the results of metallic elements in slags.

Line 393-396: I think that it should mentioned that slags can be considered as hazardous materials in a long-time scale. Your pot experiment represents short-time experiment. I think that the results of metallic elements in BOF slag are necessary in your manuscript.

General comment (rather for the future): You have the soils with relatively high content of sand (approximately 70 %). In my opinion, it is not good idea to add CaO to such soils because Ca will be leached down very quickly. In sandy soils, it is better to add CaCO3. You can discuss it in manuscript. I strongly encourage you to analyze mineral composition of slag (I.e. XRD method). It will show you which phase is the source of Ca (merwinite, prtlandite, larnite, dolomite, wollastonite, monticellite, other?) and you can discuss the susceptibility of such phase to weathering in soil conditions.

Reviewer 3 Report

About the paper with the title "Application of steel slag for degraded land remediation" I have the following comments:

From the abstract, including from the material and methods section, it is very hard to understand what data and methodologies were considered to carry out this research.

In addition, I was unable to find the main gaps in the literature addressed and the novelties of this study, because there are many works about these topics.

In my opinion the main weakness of this study is the way as the results obtained from the tests were analysed. In fact, the data obtained were assessed in a descriptive way, seeming a technical report. For the scientific paper something more robust is expected.

A consequence of this is the absence of a discussion section and the lack of a real conclusions section.

The paper is far from a publishable version.

Reviewer 4 Report

The paper is very well written and concise, within the topics of the Special Issue.
In particular I appreciated the introduction, that provides sufficient background. The methods are clearly presented, but in my opinion more details about measuring procedures could be provided (e.g. how the leachate was collected, how often the pH was measured during the 17 weeks), and literature citation of the metodology could be provided (e.g. LOI method).
I think the experimental design program is good, but as noted by the authors, the repetitions are not enough. With a number of repetitions of 2 or 3 times greater it would have been possible to carry out a robust statistical analysis (e.g. anova or similar), to validate the results obtained. On the contrary, no statistical analysis of the results was carried out in this article, therefore results are devoid of scientific value beacause it is not possible to understand if the leachate pH, the dry biomass, the germination rate, the coverage rate, the macronutrient and micronutrients concentration presents significant statistical differences among treatments. For example in a sentence the authors affirm that "The effect of slag treatment on Ca concentrations in plants was clear", but no statistical evidence is provided: is the difference significative or not?
For this reason I invite the authors to carry out at least a basic statistical analysis, which would greatly enhance their work.  

minor comments:

44 check full stop before citation [16]
44 I suggest to revise the sentence because it is not clar. e.g. "Furthemore, the production of chemical fertilizer is ..."
83 check full stop instead of comma
189 Table 6. Italics must be used for the genus and species when using Latin names of organisms, see https://www.mdpi.com/authors/layout#_bookmark15

I suggest also to check space between numbers and "%" (see the "ISO 31-0"), I remind you that superscript −1 is preferred to using “/” and to use a space between a number and its unit, see https://www.mdpi.com/authors/layout#_bookmark33

Round 2

Reviewer 3 Report

I regret but the paper remains with the weaknesses referred in my first report. 

Please address my comments of my first report. 

Reviewer 4 Report

I agree that there has been a great improvement with the addition of the t-test statistical analysis. Now the results and discussion have scientific soundness and support the conclusions. All the changes suggested by me and other reviewers were well defined, and this has greatly increased the quality of the manuscript.  For these reasons I recommend the approval of this manuscript. 

Author Response

Thank you very much for your work. We are most appreciative your comments and suggestions that have helped us to improve the quality of our manuscript.